# Design and Dynamic Modeling of a 3-RPS Compliant Parallel Robot Driven by Voice Coil Actuators

**DOI:** 10.3390/mi12121442

**Published:** 2021-11-25

**Authors:** Chuchao Wang, Shizhou Lu, Caiyi Zhang, Jun Gao, Bin Zhang, Shu Wang

**Affiliations:** School of Mechanical, Electrical & Information Engineering, Shandong University, Weihai 264209, China; 202117516@mail.sdu.edu.cn (C.W.); 201936605@mail.sdu.edu.cn (C.Z.); shdgj@sdu.edu.cn (J.G.); bin.zhang@sdu.edu.cn (B.Z.); wangshu@sdu.edu.cn (S.W.)

**Keywords:** voice coil actuator, parallel mechanism, compliant robot, dynamic model, 3-RPS

## Abstract

In order to increase the driving force of the voice coil actuator while reducing its size and mass, the structural parameters of the coil and magnet in the actuator are optimized by combing Biot–Savart law with Lagrangian interpolation. A 30 mm × 30 mm × 42 mm robot based on a 3-RPS parallel mechanism driven by voice coil actuators is designed. The Lagrangian dynamic equation of the robot is established, and the mapping relationship between the driving force and the end pose is explored. The results of dynamic analysis are simulated and verified by the ADAMS software. The mapping relationship between the input current and the end pose is concluded by taking the driving force as the intermediate variable. The robot can bear a load of 10 g. The maximum axial displacement of the robot can reach 9 mm, and the maximum pitch angle and return angle can reach 40 and 35 degrees, respectively. The robot can accomplish forward movement through vibration, and the maximum average velocity can reach 4.1 mm/s.

## 1. Introduction

Conventional rigid robots consist of high stiffness and hardness materials, which are composed of rigid elements connected by kinematic pairs [1]. Most traditional robots are actuated by motors [2]. Servo motors combined with closed-loop control enable the robots to achieve the desired motion repeatedly with high accuracy and resist a large load [3,4]. Parallel mechanisms have higher stiffness, greater payload, and more negligible inertia than their series counterparts [5]. However, the closed-loop mechanical chains reduce the working space and lower dexterity [6]. To improve their compliance, the pneumatic or hydraulic actuation substituting rigid electric motors provide flexibility for the robots benefiting from the natural fluidity and certain compressibility of the fluids. A parallel flexible rehabilitation robot actuated by four pneumatic artificial muscles (PMA) enables the ankle joints of the patients to rotate around three axes [7]. In addition, the under actuation based on variable cable length also has been widely applied in compliant parallel robots [8]. Rodriguez-Barroso et al. proposed a planar 4-DOF parallel robot with a reconfigurable end effector actuated by cables, having the advantages of great ductility, low inertia, and large working space, which can have a safe interaction with humans [9].

Although pneumatic, hydraulic, and cable actuation can provide flexibility for the robots, their actuation systems are too bulky and complex to be miniatured and integrated, limiting the applications of the robots [10]. In contrast, the voice coil actuator (VCA) has a simpler structure and smaller size than pneumatic or hydraulic actuation [11]. VCA proves to have higher power, more stiffness, and rapid response in displacement resolution [12]. It can provide a long stroke with several millimeters [13,14], which is much larger than the piezoelectric transducer (PZT) [15] and easier to control compared with the PZT [16]. VCA is the simplest linear electric motor [17], which consists of permanent magnets and coils [18]. The driving force generated by the actuator is based on the interaction between the magnetic field produced by the magnet and electrifying windings. VCA is mainly used as a linear actuator requiring high frequency, precision force, and dynamic movement [19].

Combining the parallel mechanism with the linear voice coil actuators, the representative ones are the Hexapod [20] and Stewart [21] parallel platforms used for vibration isolation benefiting from high accuracy and acceleration over a range of movement produced by voice coil actuators [22]. In order to improve the resistance capacity against the load of the platform, the size of the actuator is designed to be large, which results in high weight and low speed. A variety of optimization design methods are proposed to lower the weight of the actuator and improve its motion accuracy and dynamic performance. The Non-Dominated Sorting Genetic Algorithm-Ⅱ (NSGA-Ⅱ) [23], sequential optimization method (SOM), and dimension reduction optimization method (DROM) [24,25] are taken to minimize the dimension and mass of the cylindrical voice actuator with a certain force and stroke. The structural parameters of VCA are optimized to increase response speed [26,27], maximize the acceleration capability [28], and minimize the heat dissipation [29]. The space-mapping (SM) technique [30], manifold mapping (MM) [31], magnetic equivalent circuit method (MEC) [32], the single-phase, and the multi-phase level set method [33] are applied to maximize the output force. It is mainly to optimize the design of the VCA by improving the accuracy, increasing the speed, reducing the size, minimizing the heat dissipation, and maximizing the driving force through the above analysis. There is no consideration of the opposing relationship between the optimization indicators.

In order to increase the output force of the VCA, it is necessary to increase the volume of the coil and the magnet, but the size and mass of the VCA will rise accordingly, and the heat dissipation will also increase. In order to increase the output force while reducing the volume of the system, the structural parameters of the coil and the magnet need to be optimized. In this paper, a 3-DOF compliant robot combining voice coil actuators with a parallel mechanism was designed. In order to increase the average electromagnetic force on the magnet in the process of movement and limit the size of the VCA, the method combining Biot–Savart law with Lagrange interpolation is used to analyze the distribution of the magnetic field along the radial direction and select the structural parameters of the coil and magnet. In order to realize the control of the end pose by adjusting the input current of the coil, the dynamic analysis of the mechanism is carried out to explore the mapping relationship between the driving force and the end pose. Regarding the driving force as the intermediate variable, the mapping relationship between the input current and the motion parameters of the end pose is further obtained. The prototype of the robot is fabricated, and the experimental platform has been built to test the static and dynamic ability.

## 2. Design of the Parallel Robot

### 2.1. Mechanical Structure Design of the Robot

A 3-RPS parallel mechanism is adopted as the main structure of the robot, and its schematic is shown in Figure 1a. The robot can be divided into vertical and horizontal types according to the movement regulations and the shape of the motion platforms, as shown in Figure 1c. Limited by the overall size and weight of the structure, the spherical and revolute joints are not realized by bearings. The upper and lower plates as well as the actuators, except for the coil, magnets, and springs, are made of photosensitive resin and manufactured by 3D printing. The inner spherical hole of the upper plate is matched with the upper end of the actuator to form a spherical joint (S). The lower plate was connected with the end of the actuator through a threaded connection to form a rotating joint (R). The voice coil actuator is used as a kind of actuation and works as an indispensable part of the structure, which further realizes the integration of mechanism and actuation.

With the magnet by bonding, in order to ensure that the magnet can translate flexibly inside the support sleeve, the outer surface of the magnet has a clearance fit with the inner surface of the support sleeve. Considering the impact caused by the magnet in the process of falling, a compression spring with higher stiffness is placed at the bottom of the support sleeve to play the role of buffer and shock absorption. When the current is applied to the coil, the magnet translates approximately axially under the electromagnetic force, forming a prismatic joint (P).

The robot has three degrees of freedom, which can translate along the z-axis and rotate around the x and y-axis, as shown in Figure 1d. A variety of different motion regulations of the motion platform can be realized by controlling the input current of the three coils.

### 2.2. Theoretical Analysis of Electromagnetic Force

The electromagnetic force generated by the coil on the magnet has been analyzed. A Cartesian coordinate system is established in Figure 2a. *R*_1_ and *R*_2_ respectively represent the inner and outer radius of the coil, *l* represents the height of the coil, *I* represents the input current, and *n* represents the coil turns per unit length.

P is a selected point on the YZ plane. According to the Biot–Savart law [34], the axial component of the magnetic induction of the coil at the point P can be expressed by Equation (1).
(1)$$B_{z}(z_{0})=\frac{\mu_{0}n^{2}I}{4\pi }\int_{R_{1}}^{R_{2}}\int_{0}^{2\pi }\int_{0}^{l}\frac{\left(R^{2}-y_{0}R\sin\theta \right)}{r{\left(z_{0}\right)}^{3}}dzd\theta dR$$

$\mu_{0}=4\pi \times 10^{-7}$ H/m represents the permeability of the vacuum. The third power of the distance between points P and B can be obtained by Equation (2).
(2)$$r{\left(z_{0}\right)}^{3}={\left[R^{2}+y_{0}^{2}+{\left(z_{0}-z\right)}^{2}-2y_{0}R\sin\theta \right]}^{\frac{3}{2}}$$

As shown in Figure 2b, *z*_1_ and *z*_2_ represent the height of the upper and lower surfaces of the magnet in the established coordinate system, respectively. *r*_1_ and *r*_2_ express the inner and outer radius of the magnet, respectively. *S* represents the cross-sectional area of the magnet, and *B*_0_ represents the magnetic induction of the magnet. *B_z_*_2_ and *B_z_*_1_ represent the axial magnetic induction of the magnetic field generated by the coil on the upper and lower surfaces of the magnet, respectively. The axial electromagnetic force on the magnet can be calculated by Equation (3).
(3)$$F_{Bz}=\frac{B_{0}n^{2}I}{2}\int_{r_{1}}^{r_{2}}\int_{R_{1}}^{R_{2}}\int_{0}^{2\pi }\int_{0}^{l}\left(rR^{2}-r^{2}R\sin\theta \right)\left[\frac{1}{r\left(z_{1}\right)3}-\frac{1}{r\left(z_{2}\right)3}\right]dzd\theta dRdr$$

According to Equation (3), the axial electromagnetic force is proportional to the magnetic induction of the magnet and the input current through the coil. However, it is a definite quadruple integration, which is difficult to analyze and calculate. It is assumed that the magnetic induction at each point on each cross-section of the magnet is equal to the one at the intersection of the cross-section and the axis [35], which can transport Equation (3) into Equation (4) composed of basic elementary functions.
(4)$$F_{Bz1}=\frac{\pi B_{0}n^{2}I(r_{2}^{2}-r_{1}^{2})}{2}\left[\varphi_{B}(z_{2})-\varphi_{B}(z_{1})\right]$$

In Equation (4), a function can be defined as follows:(5)$$\varphi_{B}(z)=z\ln(\frac{R_{2}+\sqrt{R_{2}^{2}+z^{2}}}{R_{1}+\sqrt{R_{1}^{2}+z^{2}}})+(l-z)\ln(\frac{R_{2}+\sqrt{R_{2}^{2}+{\left(l-z\right)}^{2}}}{R_{1}+\sqrt{R_{1}^{2}+{\left(l-z\right)}^{2}}}).$$

Equation (4) does not consider the magnetic field at the boundary of the magnet. Therefore, Lagrangian interpolation is performed on the magnetic induction intensity of the magnet at different radii to obtain the radial distribution of the magnetic field generated by the coil. Suppose *B_zi_* represents the axial magnetic induction intensity corresponding to the different radius *r_i_* (*i* = 0, 1, … *n*). The *n*-th degree Lagrangian interpolation polynomial can be expressed as follows.
(6)$$L_{n}(r)=\sum_{i=0}^{n}\left(\prod_{j=0j\neq i}^{n}\frac{r-r_{j}}{r_{i}-r_{j}}\right)B_{zi}(r_{i})$$

The electromagnetic force can be further calculated by Equation (7).
(7)$$F_{Bz2}=\frac{2\pi B_{0}}{\mu_{0}}\int_{r_{1}}^{r_{2}}\left[rL_{n}(r){|}_{z=z_{2}}-rL_{n}(r){|}_{z=z_{2}}\right]dr$$

### 2.3. Parameter Selection of the Actuator

Limited by the overall diameter and height of the robot, the outer radius of the coil is 4 mm, the height is 15 mm, and the diameter of the copper wire is 0.2 mm. A solid magnetic column with a height of 18 mm is selected. Set the outer diameter and the inner diameter of the coil varying from 0.25 to 3.5 mm, and the size interval is set to 0.25 mm. The displacement of the magnet varies from 0 to 9 mm, and the interval is set to 0.1 mm. The average electromagnetic force on the magnet in the process of translation is analyzed based on different sizes of the magnet and coil when the input current is set to 1A.

For each simulation parameter above, ANSYS Maxwell software is used to analyze the electromagnetic force on the magnet at different displacements, and the average value is obtained as the average electromagnetic force *F_Bz_*_3_ received by the magnet through simulation under this parameter. Similarly, the average electromagnetic forces *F_Bz_*_1_ and *F_Bz_*_2_ can be calculated by Equations (4) and (7). The flow chart of the analysis is shown in Figure 3.

The relationship between the average electromagnetic force and size parameters is obtained by Equations (4) and (7) (*n* = 4) and simulation, as shown in Figure 4a–c, respectively. It can be seen that the average electromagnetic force increases with the decrease in the gap between the coil and magnet when the outer diameter of the magnet is kept constant. The magnetic permeability of the NdFeB magnet is much higher than that of the air. On the other hand, the average electromagnetic force increases first and then decreases with the rise of the outer diameter of the magnet when the gap remains fixed. The values of the outer radius for the magnet and the gap are used for linear fitting when the electromagnetic force reaches the maximum. Within a 95% confidence interval, the results obtained in different methods are shown in Figure 4d–f.

According to the above analysis, the air gap should be reduced as much as possible to improve the electromagnetic force. The outer diameter of the magnet and the inner diameter of the coil are further obtained through the fitting curve after the determination of the gap between them. It can be seen that the maximum error of the fitting value is less than 0.25 mm from the comparison of several fitting curves. The value of electromagnetic force calculated by Equation (7) compared with that calculated by Equation (5) is closer to that received through simulation. The air gap is set to 1 mm, considering the thickness of the support sleeve. Combined with the fitting values and actual requirements, the size and material of the components in the actuator are shown in Table 1. The performance parameters of the actuator are shown in Table A1.

## 3. Dynamic Modeling of the Robot

The global static coordinate system {*O*_0_-*X*_0_*Y*_0_*Z*_0_} is established at the geometric center of the triangle *R*_1_*R*_2_*R*_3_, the dynamic coordinate system {*O*_7_-*X*_7_*Y*_7_*Z*_7_} of the motion platform is established at the geometric center of the triangle *S*_1_*S*_2_*S*_3_, and the local coordinate system is established on the corresponding link, as shown in Figure 5. The length of *O*_0_*R_i_* and *O*_7_*S_i_* (*i* = 1,2,3) is represented by *r.* The position vector of the dynamic coordinate system {*O*_7_} relative to the static coordinate system {*O*_0_} is described as: $\vec{P}={\left[x_{0},\;\;y_{0},\;\;z_{0}\right]}^{T}$. The xyz Euler angles are used to describe the orientation of {*O*_0_} relative to {*O*_7_}.

### 3.1. Dynamic Modeling with the Euler–Lagrangian Approach

By establishing the Lagrangian dynamic equation of the parallel mechanism, the inverse dynamic analysis is carried out to explore the mapping relationship between the pose of the motion platform and the driving force. The driving force *F_i_* (*i* = 1,2,3) generated by the voice coil actuator is selected as the generalized force, and the displacement *d_i_* (*i* = 1,2,3) of the magnet is selected as the generalized coordinate. The general expression of the Lagrangian equation under the condition of no external load is obtained by Equation (8).
(8)$$\frac{d}{dt}\frac{\partial L}{\partial \overset{\cdot }{q}}-\frac{\partial L}{\partial q}=Q$$

*Q* represents the generalized force corresponding to the generalized coordinate *q*, and the Lagrangian operator can be expressed by Equation (9).
(9)L=T−U

Corresponding to the local coordinate system established in Figure 5, ^0^*A**_i_* (*i* = 1–7) respectively represents the transformation matrix of {*O**_i_*} relative to the global static coordinate system. The kinetic energy of the parallel mechanism can be described by Equation (10).
(10)$$T=\frac{1}{2}\sum_{j=1}^{3}\sum_{k=1}^{3}\left[\sum_{i=1}^{7}tr\left(\frac{\partial^{0}A_{i}}{\partial q_{j}}J_{i}\frac{\partial {(}^{0}A_{i}{)}^{T}}{\partial q_{k}}\right)\right]{\overset{\cdot }{q}}_{j}{\overset{\cdot }{q}}_{k}$$

Assuming that *g*_0_ represents the gravitational acceleration in the global coordinate system. *^i^r_ci_* (*i* = 1–7) represents the vector coordinate of the centroid in the local coordinate system of the link *i*. The gravitational potential energy of the parallel mechanism can be described by Equation (11).
(11)U=−∑i=17mig0→Ai0rcii→

In Equation (10), *J_i_* represents a constant matrix describing the mass distribution of link *i* in its local coordinate system. *^i^I_x_*, ^*i*^*I_y_*, and *^i^I_z_* represent the moment of inertia about link *i* with respect to three coordinates axes. *^i^I_xy_*, *^i^I_yz_*, and *^i^I_zx_* represent the product of inertia about the link *i*. *^i^x_ci_*, *^i^y_ci_*, and *^i^z_ci_* represent the coordinate components of the mass center in link *i* along three axes in the local coordinate system. *m*_i_ represents the mass of link *i*. *J_i_* (*i* = 1–7) can be calculated by Equation (12).
(12)Ji=[−Ixi+Iyi+Izi2IxyiIxzimixciiIxyiIxi+Iyi+Izi2IyzimiyciiIxziIyziIxi+Iyi+Izi2mizciimixciimiyciimizciimi]

By substituting Equations (9)–(12) to Equation (8), the Lagrangian equation can be transformed into Equation (13).
(13)$$\left[\begin{array}{l} h_{11}\;h_{12}\;h_{13}\\ h_{21}\;h_{22}\;h_{23}\\ h_{31}\;h_{32}\;h_{33} \end{array}\right]\overset{\cdot \cdot }{q}+\left[\begin{array}{l} \dot{q^{T}}C_{1}\dot{q}\\ \dot{q^{T}}C_{2}\dot{q}\\ \dot{q^{T}}C_{3}\dot{q} \end{array}\right]\overset{\cdot }{q}+\left[\begin{array}{l} g_{1}\\ g_{2}\\ g_{3} \end{array}\right]=\left[\begin{array}{l} \tau_{1}\\ \tau_{2}\\ \tau_{3} \end{array}\right]$$

In Equation (13), *h_jk_*, *C_j_*, and *g_j_* (*j*,*k* = 1,2,3) can be calculated by Equations (14)–(16).
(14)$$h_{jk}=\sum_{i=1}^{7}tr\left(\frac{\partial^{0}A_{i}}{\partial q_{j}}J_{i}\frac{\partial {(}^{0}A_{i}{)}^{T}}{\partial q_{k}}\right)$$
(15)$$C_{j}=\sum_{k=1}^{3}\sum_{i=1}^{3}\left(\frac{\partial h_{jk}}{\partial q_{i}}-\frac{1}{2}\frac{\partial h_{ki}}{\partial q_{j}}\right)$$
(16)gj=∑i=1nmig0∂Ai0∂qjrcii

The driving force generated by each actuator can be obtained by substituting Equations (14)–(16) into Equation (13). It is assumed that *k_j_* (*j* = 1,2,3) represents the stiffness coefficient of the spring in the actuator and *f_j_* (*j* = 1,2,3) represents the damping coefficient. Then, the axial electromagnetic force *F_BZj_* (*j* = 1,2,3) can be calculated by Equation (17).
(17)$$F_{Bzj}=\tau_{j}-k_{j}d_{j}-f_{j}{\overset{\cdot }{d}}_{j}$$

The current *I_j_* (*j* = 1,2,3) can be further calculated, according to Equation (7). The flow chart of the inverse dynamic algorithm when programming with MATLAB software is shown in Figure 6.

### 3.2. Dynamic Simulation Analysis

The dynamic simulation analysis of the robot is carried out to verify the results from the inverse dynamic analysis on the dynamic simulation software ADAMS. Firstly, import the 3D model of the robot and add materials to each part, as shown in Figure 7a. Secondly, add corresponding constraints to the model, as shown in Figure 7b. According to the expected motion trajectory of the robot, the mapping relationship between the driving force and time can be solved through the above inverse dynamic algorithm and imported into ADAMS, as shown in Figure 7c. Then, the springs and dampers are added at the appropriate locations inside the actuators, as shown in Figure 7d. The stiffness coefficients of the springs and the damping coefficients of the dampers are set to 9.9 N/m and 2 N.s/m, respectively. Driving force and gravity are added to the model, as shown in Figure 7e. Finally, after the pre-processing is completed, the simulation analysis is carried out, and the solution results are exported, as shown in Figure 7f.

For example, suppose the motion platform translates along the Z-axis while rotating around the X-axis. The motion trajectories are z = 3 − 3cos(*t*) and α = 2.5 − 2.5cos(*t*), respectively. According to the inverse dynamic algorithm described in Figure 6, the electromagnetic force and input current varying with time are shown in Figure 8c,f. The expected motion regulations are compared with the simulation results, as shown in Figure 8a,b,d,e. The maximum error in the displacement along the Z-axis is less than 0.2 mm. A sudden change exists in the curve that speed varies with time at the initial moment because of the spring and damping set during the simulation. Except for that, there is an error less than 0.05 mm in the curve of velocity along the Z-axis, rotation angle, and angular velocity around the X-axis.

The fitting results of the mapping relationship on electromagnetic force and current changing with time can be expressed as the sum of two sine functions within a 95% confidence interval on the MATLAB software, corresponding to the two sinusoidal movements of the motion platform. Assume that they can be expressed as Equation (18). The fitting coefficients of the electromagnetic forces and currents are shown in Table 2.
(18)f(t)=asin(bt+c)+dsin(et+f)

## 4. Experiments of the Robots

Based on the above analysis of the robot, the static and dynamic performances of the robot have been tested. An experimental platform has been built, which can be divided into three layers, as shown in Figure 9. The first floor includes power and display parts; 18650 type Li-ion batteries provide power for the whole system. Three DC voltammeters display the voltage and input current through the coils in the three actuators. The second floor constitutes the control system. An STC89C52 microcontroller is used as the main control chip. An LM2596 voltage regulator module is applied to regulate the voltage supplied by the batteries. Two L298n modules utilize PWM signals generated by the microcontroller to modulate the output voltage supplied by the LM2596. By changing the duty cycle of the PWM signal, the current through the coils can be further varied. The third floor consists of two robots for the static performance test, which includes the load resistance test and the rotation test of the motion platform.

### 4.1. The Static Performance Test of the Robot

#### 4.1.1. The Load Resistance Test of the Robot

When the robot is in equilibrium, the axial displacement varying with input voltages under different loads has been explored. As shown in Figure 10a, loads are simulated by different weights, the input voltages are altered by changing the duty cycle of the PWM signal, and the axial displacement is measured by the ruler. The standard error is 0.5 mm. In order to make the motion platform of the robot translate axially, the duty cycles of the PWM signals corresponding to the voltage through three coils are kept the same. The average value of the voltages through the three coils is used as the input voltage of the robot.

Firstly, the axial displacement varying with the voltage is measured under the condition of no load. Then, we place different weights varying from 1 to 10 g and record the voltage and the displacement. The results are shown in Figure 10b and Table A2. The experiment is repeated eleven times. When the load is 0 g and 1 g, the axial displacement gradually increases to the limit height with the rise of current. When the load varies from 2 to 10 g and the initial input voltage is small, the axial displacement remains 0 mm because the output forces of the actuators are less than the external load, and then, the displacement gradually increases to the limit height with the increase in voltage. When the input voltage becomes larger, greater output forces generated by the actuators than the load enable the motion platform to maintain the maximum displacement. The area where the displacement remains constant with the change of voltage is called a static area. When the input voltage is kept the same, the displacement decreases with the increase in load on the whole.

Corresponding to different input voltages, the fitting curves of the applied load and axial displacement based on a fourth-order polynomial function are shown in Figure 11. The fitting values and the slopes at different points are calculated. The slopes of curves representing stiffness firstly increase and then decrease with the rise of displacement, as shown in Figure 11c,d,f, which explains the generation of the static area. The stiffness will change accordingly with the variation of input voltage when the axial displacement remains the same. In summary, the output stiffness can be adjusted by changing the displacement and input voltage for the electromagnetic system.

#### 4.1.2. The Rotation Ability Test of the Robot

In the practical application of the robot, the rotation angles of the motion platform represent vital indexes to measure its performance. Under different modes of actuation, α and β of the motion platform rotating around the X-axis and Y-axis have been measured and analyzed with different input currents. As shown in Figure 12a, a JY-62 six-axis attitude angle sensor is embedded in the motion platform, which can communicate with upper computer software through the serial port to measure and transport the rotation angles of the motion platform around the three axes.

Theoretically, there is only β rotating around the Y-axis under a single actuator, as shown in Figure 12b. In this actuation, the measurement of angles with the current is shown in Figure 13a. With the rise of the current, β gradually increases to a maximum of 40 degrees. The values of α and γ are not all zero, but they are much smaller than that of β. When the polarities of the current are changed, the mode of the two actuators is taken, as shown in Figure 12c. The variation of β changing with the current has been shown in Figure 13b. As the current increases, the β also increases to a maximum of 40 degrees. When the input current is small, there is a more significant error in α due to the length error of two actuators. When another electromagnetic actuator is selected for driving, as shown in Figure 12d, γ is much smaller than α and β. With the increase in current, α gradually increases to 35 degrees, as shown in Figure 13c. Similarly, the angles varying with current are shown in Figure 13d when the polarities of current are changed, as shown in Figure 12e. The experimental results of the rotation ability test are shown in Table A3.

### 4.2. The Dynamic Performance Test of the Robot

Snails, a common invertebrate soft creature, depend on their wide abdominal pleopod interacting with the ground to produce wave-like forward movement [36,37]. As shown in Figure 14, a square wave voltage is applied to the one coil of the robot so that the motion platform can produce periodic swings. The lower plate of the robot further interacts with the ground so that the robot can move forward depending on the friction and changes in the center of gravity to imitate the movement of natural organisms.

The moving speed of the robot can be changed by adjusting the amplitude and frequency of the voltage signal. When the amplitude and frequency are low, the vibration and frequency of the robot and input electrical energy are so slight that the robot cannot move forward. On the contrary, when the amplitude and frequency are too high, the movement of the robot will be uncertain, so the joint cannot keep moving forward. Therefore, the duty cycle of the PWM signal is set to 30%, 40%, and 50%, and the frequency of the voltage signal is set to 10 Hz, 12 Hz, and 15 Hz, respectively. The maximum movement distance is set to 170 mm.

The average speed firstly increases and then decreases with the increase in voltage frequency when the duty cycle of the PWM signal remains constant, as shown in Figure 15a–c. The robot can move by the periodic forward force caused by the interaction between the lower plate and the surface of the platform. The robot moves faster with the rise of frequency, which means that the number of interactions per second increases. The forward movement occurs during the extension of the actuator through observation, and the input square wave voltage has the same forward and reverses duration. When the frequency continues to increase and the voltage period is less than the motion period, the duration of the driving force is not enough to maintain the movement of the robot, so the moving speed gradually decreases. Through the above analysis, the input excitation frequency does not match the output motion frequency, which is the cause of the above phenomenon. When the vibration frequency is much greater than the movement frequency, the robot will have an uncertain direction of movement or even vibrate in place. As shown in Figure 15d–f, increasing the duty cycle of the PWM signal will provide more energy to the system and increase the speed of the robot when the frequency remains constant. The results representing the average speed of the robot at different frequencies and duty cycles are shown in Table A4.

## 5. Conclusions

A compliant robot driven by voice coil actuators has been designed. Through the analysis of the electromagnetic force, the structural parameters of the coil and magnet are determined. The kinematics and dynamics of the 3-RPS mechanism are analyzed. The static and dynamic performances of the robot are tested. The main conclusions are as follows.

(1)The electromagnetic force calculated by Biot–Savart law combined with Lagrangian interpolation takes into account the magnetic field at the boundary point of the magnet, and the values are closer to simulation results. When the electromagnetic force reaches the maximum, the outer diameter of the magnet changes linearly with the gap within the 95% confidence interval, which is used as the regulation for the selection of structural parameters.(2)The input current and the pose of the motion platform are regarded as the input and output, respectively. The current can be further transformed into the motion parameters of the robot through dynamic analysis by taking the driving force as the intermediate variable. The end pose of the motion platform can be adjusted by regulating the input current through the mapping relationship between them.(3)The robot can accomplish basic movements such as axial translation and the rotation of the motion platform. The output stiffness can be regulated by varying the input voltage and the displacement of the robot. In addition, the robot can move forward relying on the vibration. With the increase in the input frequency, the speed of the joint firstly increases and then decreases when the input voltage is kept constant. With the rise of the input voltage amplitude, the speed of the robot increases when the input voltage keeps constant.

The current work has the following problems that need to be improved. The static and dynamic performances of the robot are tested by using the open-loop control. At present, only physical cooling methods are used to reduce the coil heat dissipation.

In the future, our work will focus on the establishment of an integrated system model of the structure, actuation and sensing as well as the force position control of the robot. At present, we use a six-axis attitude angle sensor to detect the orientation of the robot. Rigid sensors have poor flexibility, large inertia, and are not suitable for the integration of the system. In our future work, we will take advantage of soft sensors, such as the DE sensor, to detect the displacement of the actuator. Taking into account the accuracy of the end pose and the ability to interact with the external environment, it is necessary to take control of the force position of the robot, such as the force–position coupling control, impedance control, and adaptive control.

## Figures and Tables

**Figure 1 micromachines-12-01442-f001:**
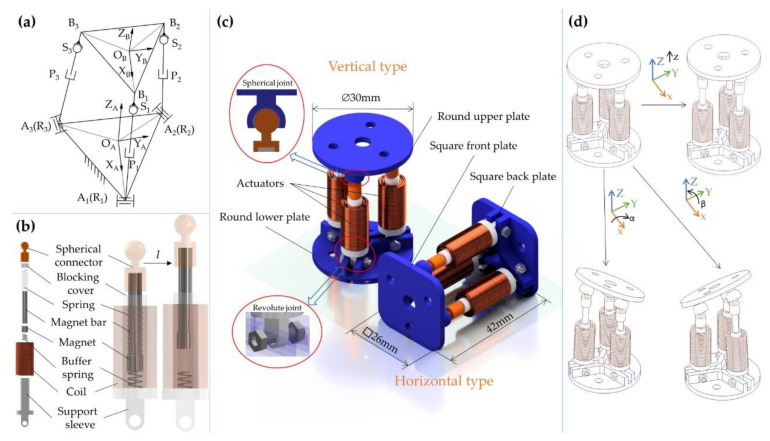
Structure of the robot. (**a**) Schematic model of the 3-RPS parallel mechanism. (**b**) Structure of the voice coil actuator. (**c**) CAD model of the robots. (**d**) Movement of the mobile platform.

**Figure 2 micromachines-12-01442-f002:**
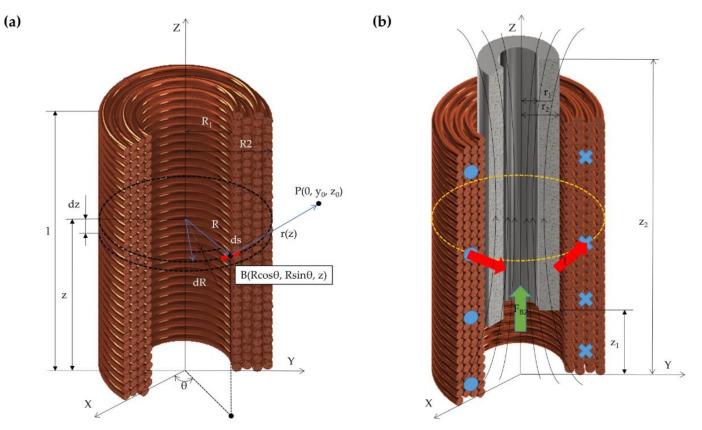
Analysis and calculation of the electromagnetic force. (**a**) Schematic of the magnet field generated by the electromagnetic coil at one point. (**b**) Diagram of axial electromagnetic force generated by the coil on the coaxial magnet.

**Figure 3 micromachines-12-01442-f003:**
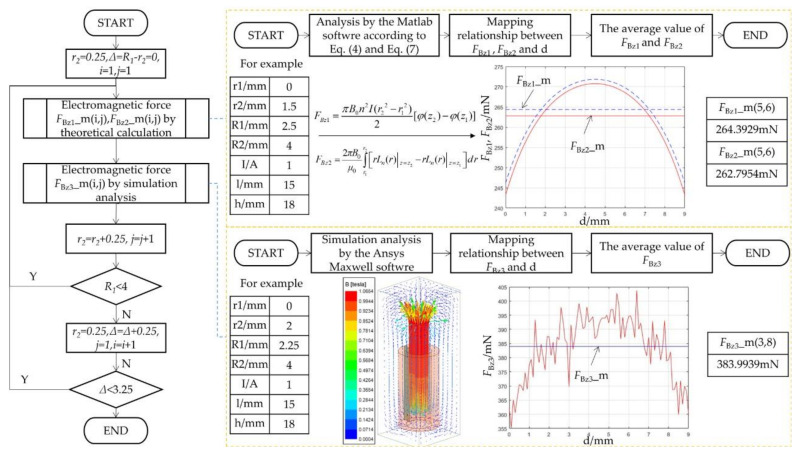
The flow chart of the average electromagnetic force received by the magnet.

**Figure 4 micromachines-12-01442-f004:**
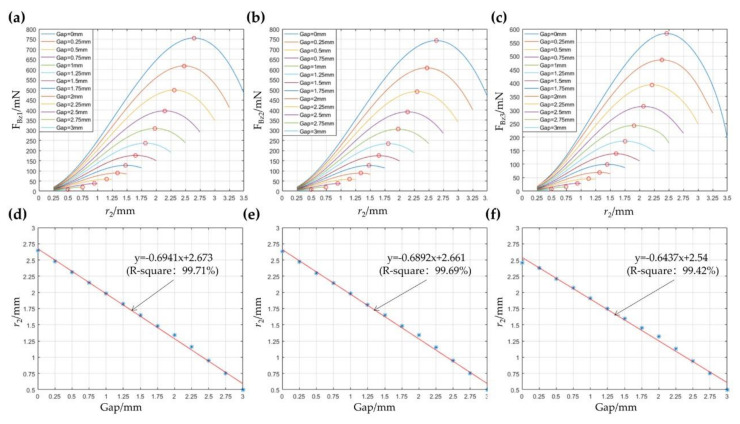
Diagrams of average electromagnetic forces varying with simulation parameters through Equation (4) (**a**), Equation (7) (**b**), and simulation analysis (**c**). The results of the linear fitting through Equation (4) (**d**), Equation (7) (**e**), and simulation analysis (**f**).

**Figure 5 micromachines-12-01442-f005:**
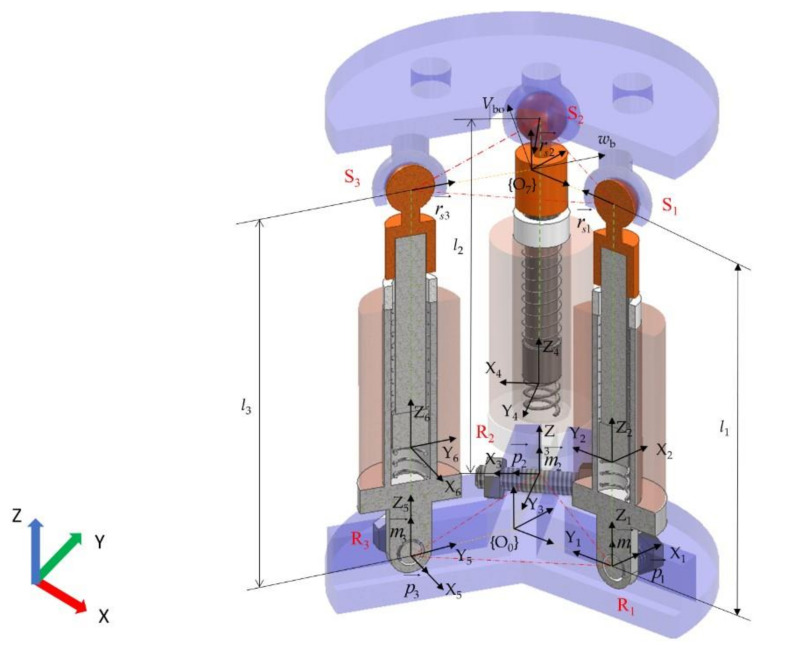
Establishment of the coordinate system for the robot.

**Figure 6 micromachines-12-01442-f006:**
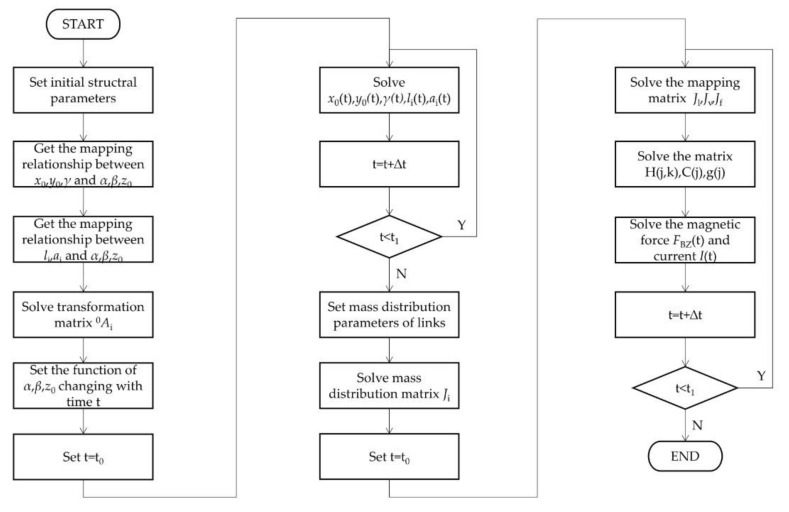
The flow chart of the inverse dynamic algorithm.

**Figure 7 micromachines-12-01442-f007:**
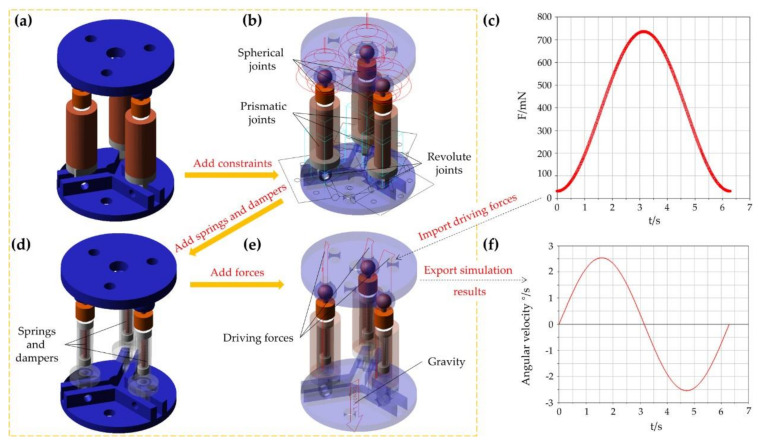
Dynamic simulation analysis of the robot. (**a**) Import 3D model. (**b**) Add corresponding motion joints. (**c**) Import external driving force. (**d**) Add the springs and dampers. (**e**) Add the actuation and gravity. (**f**) Export the simulation results.

**Figure 8 micromachines-12-01442-f008:**
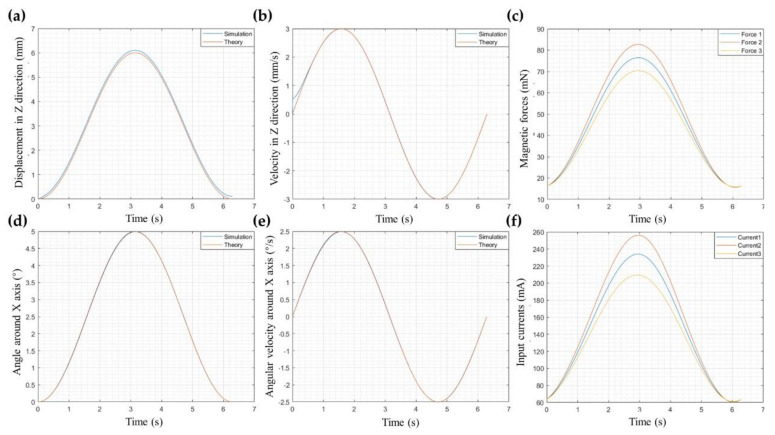
Comparison of displacement (**a**) and velocity (**b**) in the Z-direction between theoretical analysis and simulation. (**c**) The magnetic forces changing with time. Comparison of rotating angle (**d**) and angular velocity (**e**) around the X-axis between theoretical calculation and simulation. (**f**) The input current changing with time.

**Figure 9 micromachines-12-01442-f009:**
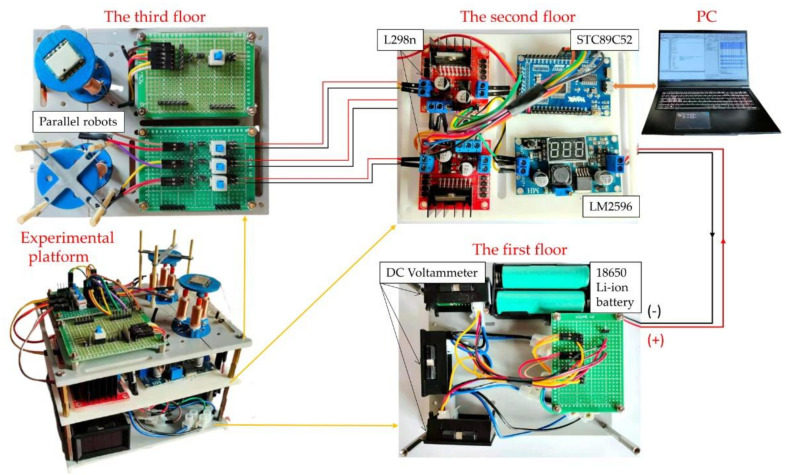
An experimental platform for testing the performance of the robots.

**Figure 10 micromachines-12-01442-f010:**
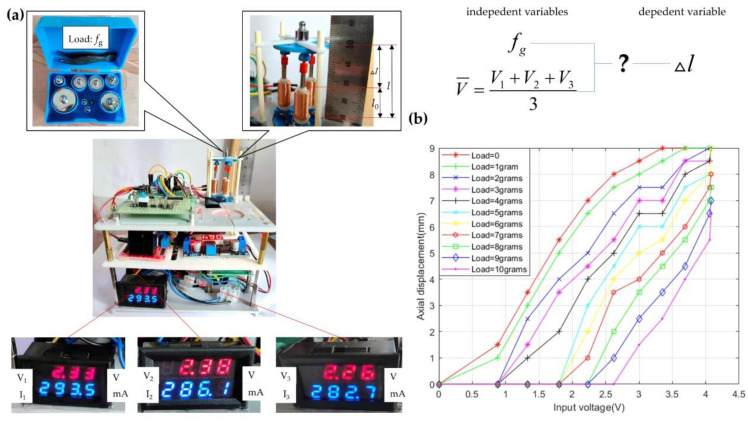
The process and results of the load resistance test. (**a**) The equipment for the test. (**b**) The curves of axial displacement of varying with loads and input voltages.

**Figure 11 micromachines-12-01442-f011:**
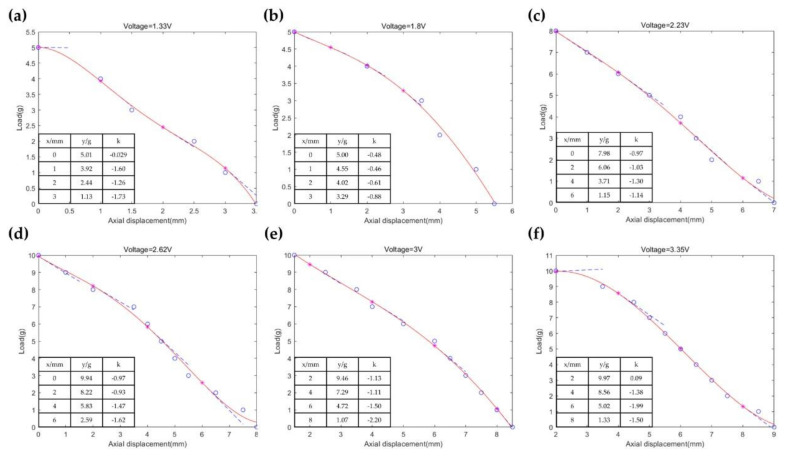
The fitting results of load over axial displacement when the voltage is 1.33 V (**a**), 1.8 V (**b**), 2.23 V (**c**), 2.62 V (**d**), 3 V (**e**), and 3.35 V (**f**).

**Figure 12 micromachines-12-01442-f012:**
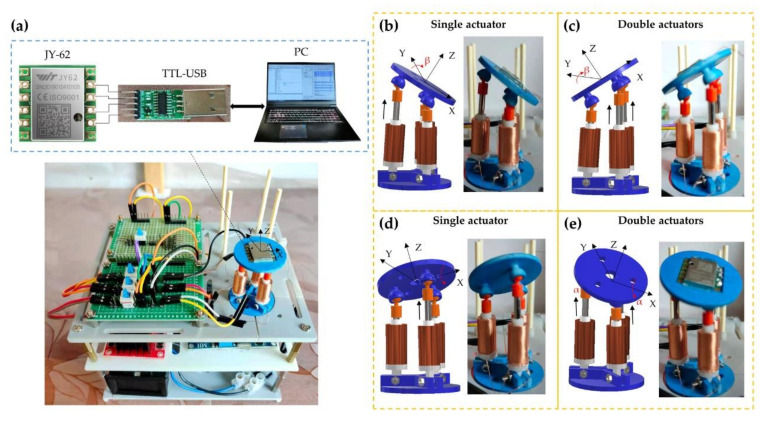
Rotatory ability test of the robot. (**a**) The equipment for the test. Different modes of actuation have been shown in (**b**–**e**).

**Figure 13 micromachines-12-01442-f013:**
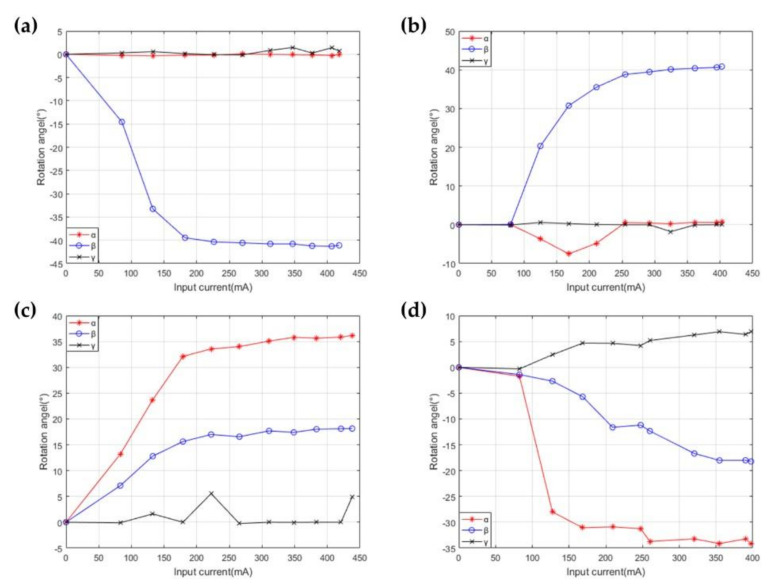
Variation of angles with the current. (**a**) The test of measuring β in the mode of a single actuator. (**b**) The test of measuring β in the mode of double actuators. (**c**) The test of measuring α in the mode of a single actuator. (**d**) The test of measuring α in the mode of double actuators.

**Figure 14 micromachines-12-01442-f014:**
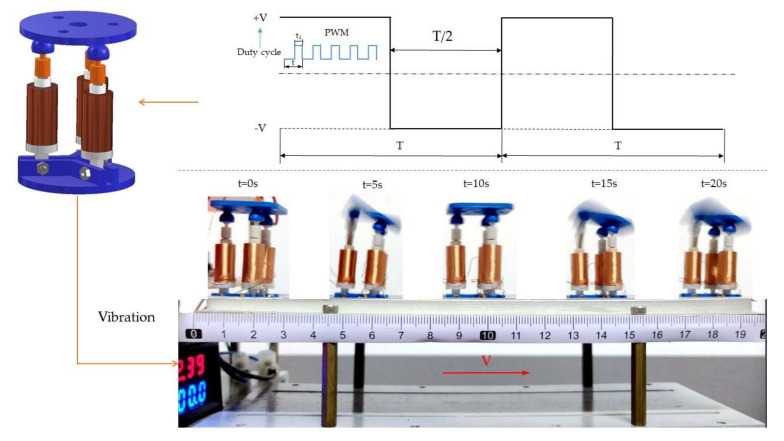
The forward movement of the robot through vibration.

**Figure 15 micromachines-12-01442-f015:**
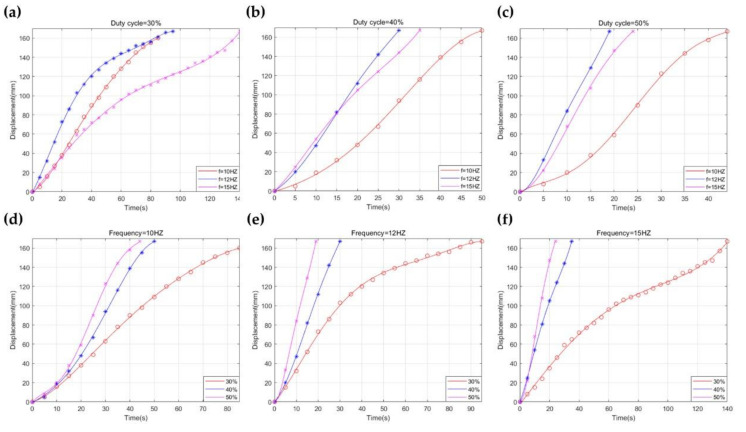
The moving displacement varying with the time, when the duty cycle of the PWM signal is kept at 30% (**a**), 40% (**b**), 50% (**c**), and the input frequency receives 10 Hz (**d**), 12 Hz (**e**), and 15 Hz (**f**).

**Table 1 micromachines-12-01442-t001:** Parameter table of components in the actuator.

Component	Parameter/mm	Value	Material
Coil	Inner diameter	4.5	Copper
	Outer diameter	8	
	Height Wire diameter	15 0.2	
Magnet bar	Outer diameter	2.5	NdFeB (N53)
	Height	12	
Soft spring	Outer diameter	3	SUS304
	Height Wire diameter	9 0.1	
Magnet	Outer diameter	3	NdFeB (N53)
	Height	3	
Buffer spring	Outer diameter	2.8	SUS304
	Height Wire diameter	3 0.3	

**Table 2 micromachines-12-01442-t002:** The fitting parameters of the forces and currents.

	a	b	c	d	e	f	R-Square
*F* _1_	46.04	0.006	1.553	30.29	1	4.912	100%
*F* _2_	49.39	0.002	1.560	33.72	1	4.912	100%
*F* _3_	42.73	0.022	1.505	26.81	1.001	4.910	100%
*I* _1_	158.6	0.171	1.099	74.70	1.058	4.752	99.98%
*I* _2_	180.1	0.234	0.906	74.87	1.108	4.598	99.98%
*I* _3_	158.8	0.005	2.102	73.76	1	4.951	99.98%

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
