# Peer review of "Design and Dynamic Modeling of a 3-RPS Compliant Parallel Robot Driven by Voice Coil Actuators"

_micromachines, 2021, doi:10.3390/mi12121442_

Round 1

Reviewer 1 Report

The authors present the article entitled “Design and Dynamic Modeling of a 3-RPS Compliant Parallel Robot Driven by Voice Coil Actuators”. The article is interesting and is easy to read. However, it presents the following concerns:

Line 28: Please correct the citation style: it is incorrect to write

 Rodriguez-Barroso et al. [5]

Lines 58-68: The objective of the manuscript is not clear. Please explain it clearly and highlight the novelty of the work.

Add future works in the conclusion section

Remove section 6 headline.

Please, discuss the advantages of traditional robotic architectures seen in up-to-date references as follows in order to highlight your contribution

An FPGA-based open architecture industrial robot controller

A multidisciplinary industrial robot approach for teaching mechatronics-related courses

Concurrent optimization for selection and control of ac servomotors on the powertrain of industrial robots

A PID-type fuzzy logic controller-based approach for motion control applications

Please, improve the quality of the images, I recommend you vectorize them (if you are working in Word, you can use EMF format). In this sense, the graphs can be detailed in a better way.

Reviewer 2 Report

The article describes the design and dynamic analysis and verification of robot properties with parallel kinematic structure. Voice coil actuators were chosen to drive the robot. The article is written in a good scientific style, but I recommend:
- edit the abstract to better describe the focus of the article
- add in the introduction why the authors focused on this problem
- write the findings from the experiment in Figure 14,15
- In Conclucion add how the research will continue
- check chapter 6 Patents

The article is very interesting for the readers and very well graphically processed. Mathematical processing is at a good level and sufficiently describes the problem. I did not reveal plagiarism in the article and I did not reveal conflict in the references. After minor revision, I recommend publishing the manuscript.

Reviewer 3 Report

Brief Description of the Work

The main objective of the work is to increase the average electromagnetic force on the magnet in the process of movement. To this aim, the authors fabricated a prototype of robot based on 3-RPS parallel mechanism driven by voice coil actuators. The distribution of magnetic field along the radial direction and the structural parameters of the coil and magnet have been studied by means of the Biot-Savart law combined with Lagrange interpolation. The electromagnetic force calculated by this method takes into account the magnetic field at the boundary point of the magnet. The input current and the pose of the motion platform are regarded as the input and output, respectively. The static and dynamic performances of the robot have been tested by building an experimental platform.

Main Results Obtained

Main performances of the robot

- The robot can accomplish basic movements such as axial translation and the rotation of the motion platform;

- The robot can move forward relying on the vibration;

Technical characteristic of the robot

- The robot can bear a load of 10 g;

- The maximum axial displacement of the robot can reach 9 mm;

- The maximum return angle and pitch angle can reach 40 and 35 degrees, respectively.

General Consideration (GC)

GC1) The manuscript is well written and very interesting.

GC2) I did not find mathematical mistakes.

GC3) The authors describe the characteristics of the robot prototype very well, showing its advantages. However, they do not mention the limitations of the robot and what the possible drawbacks may be.

There are a couple of issues that, in my opinion, need to be clarified.

Questions/Suggestions (Q/S)

Q/S1) Motion quality of the proposed prototype robot.

For clarity, please report, maybe by using a sort of summary table:

Resolution, Accuracy, Speed, Temperature stability of the actuator

Q/S2) Classes of actuators can be compared in a quantitative manner using actuator performance indices. Examples of actuator performance characteristics include maximum actuation stress, maximum actuation strain, actuator mass density, and minimum strain resolution. Other performance characteristics include actuator modulus, maximum operation frequency, maximum power density, and power efficiency. May the authors provide the summary actuator performance indices for their actuator?

Q/S3) It would be very interesting to have a comparison between the authors' electromagnetic actuator and piezoelectric actuators. In order to have an idea about the real improvement of authors' robot performance, is it possible have a table comparing the properties between the prototype proposed by the authors and a reference Piezo-type actuator (chosen by the authors). More specifically, the table should compare:

Actuator stiffness, Amplification mechanism torsional stiffness, Actuator moving mass, Payload mass, Amplification mechanism total mass, Nominal actuator stroke, Desired system motion range.

Q/S4) Moving magnet actuators are generally considered deemed to be best suited for achieving large range nano-positioning compared to Voice coil actuators. With reference to their prototype, what is the opinion of the authors about it?

Q/S5) As mentioned above, the authors do not mention the major drawbacks of their robot prototype. This question is intended to bridge this gap.

Q/S5a) As known, the main advantages of Voice coil actuators (VCAs) are: high precision/resolution, large travel range, and high speed. Do the authors confirm that these (positive) characteristics are maintained by their robot prototype?

Q/S5b) As known, the main disadvantages of VCAs are: moving wires, heat dissipation at mover, poor thermal management, high heat output, difficult coil cooling. The authors are asked to mention which of these disadvantages were solved by their robot prototype, producing comments about this.

Conclusions

As mentioned, the work is very interesting and I really enjoyed reading it. It describes in an exhaustive way the authors' robot prototype. I think that it deserves to be published. However, I think that the authors have not sufficiently described the main advantages of their prototype over traditional type of actuators, and, in particular, they did not mention which disadvantages remain unsolved. I therefore encourage the authors to take into account the suggestions reported above.

Round 2

Reviewer 3 Report

The authors answered satisfactorily, point by point, to the questions/suggestions mentioned in my first report. I confirm that the work is very interesting and it was a pleasure to read it. I wish the authors all the best in their future research.